# Pluronic F-127 Hydrogels Containing Copper Oxide Nanoparticles and a Nitric Oxide Donor to Treat Skin Cancer

**DOI:** 10.3390/pharmaceutics15071971

**Published:** 2023-07-18

**Authors:** Fernanda V. Cabral, Bianca de Melo Santana, Camila N. Lange, Bruno L. Batista, Amedea B. Seabra, Martha S. Ribeiro

**Affiliations:** 1Center for Lasers and Applications, Nuclear and Energy Research Institute (IPEN-CNEN), São Paulo 05508-000, SP, Brazil; fe_vcabral@hotmail.com; 2Center for Natural and Human Sciences (CCNH), Federal University of ABC (UFABC), Santo André 09210-580, SP, Brazil; b.melo@aluno.ufabc.edu.br (B.d.M.S.); camilosalange@yahoo.com (C.N.L.); bruno.lemos@ufabc.edu.br (B.L.B.)

**Keywords:** CuO nanoparticles, drug delivery, melanoma, reactive oxygen species, S-nitrosoglutathione

## Abstract

Melanoma is a serious and aggressive type of skin cancer with growing incidence, and it is the leading cause of death among those affected by this disease. Although surgical resection has been employed as a first-line treatment for the early stages of the tumor, noninvasive topical treatments might represent an alternative option. However, they can be irritating to the skin and result in undesirable side effects. In this context, the potential of topical polymeric hydrogels has been investigated for biomedical applications to overcome current limitations. Due to their biocompatible properties, hydrogels have been considered ideal candidates to improve local therapy and promote wound repair. Moreover, drug combinations incorporated into the polymeric-based matrix have emerged as a promising approach to improve the efficacy of cancer therapy, making them suitable vehicles for drug delivery. In this work, we demonstrate the synthesis and characterization of Pluronic F-127 hydrogels (PL) containing the nitric oxide donor S-nitrosoglutathione (GSNO) and copper oxide nanoparticles (CuO NPs) against melanoma cells. Individually applied NO donor or metallic oxide nanoparticles have been widely explored against various types of cancer with encouraging results. This is the first report to assess the potential and possible underlying mechanisms of action of PL containing both NO donor and CuO NPs toward cancer cells. We found that PL + GSNO + CuO NPs significantly reduced cell viability and greatly increased the levels of reactive oxygen species. In addition, this novel platform had a huge impact on different organelles, thus triggering cell death by inducing nuclear changes, a loss of mitochondrial membrane potential, and lipid peroxidation. Thus, GSNO and CuO NPs incorporated into PL hydrogels might find important applications in the treatment of skin cancer.

## 1. Introduction

Malignant melanoma is one of the most common and serious types of skin cancer worldwide [1]. It arises from the transformation of melanocytes to melanoma in the epidermis, and it is often caused by the long-term exposure of skin to ultraviolet radiation [1,2]. It is the leading cause of death from skin cancer, accounting for over 80% of deaths in those individuals affected by this disease [3]. It has been estimated that over 350,000 people were affected by melanoma in 2020, with nearly 57,000 deaths across the globe [4]. In fact, the number of new cases is expected to increase in the next years, thus impacting over 510,000 people, with approximately 96,000 deaths by 2040 [4,5].

The treatment and management of melanoma may vary with its stage, which is classified depending on the tumor thickness, ulceration, depth, nodal status, and the presence of metastasis [1,6]. The primary treatment for the early stage of the disease is the complete resection of the tumor with a wide surgical margin [1]. However, disease relapses are quite usual [6]. Therefore, in cases of local recurrences or intralymphatic metastasis, surgical excision combined with intralesional/topical therapies, or even radiotherapy, is highly recommended [1,6]. In more severe cases, systemic therapies are employed for those with inoperable tumors or presenting distant metastasis [1].

Although local treatment is effective, it can irritate the skin, leading to severe local reactions [7], which demands the development of new approaches to minimize drug toxicity. In this context, hydrogels have attracted great attention over the last few years as they can act like local matrices, thus contributing to tissue repair and wound healing [8]. Particularly, Pluronic F-127 (PL) hydrogels are suitable for biomedical applications, once they are biocompatible, hence being attractive candidates for wound dressings [9]. In addition, they have shown great potential for therapeutic applications offering many advantages as vehicles in drug delivery, mainly in topical applications [10,11]. In fact, it has been demonstrated that PL hydrogels are very effective matrices against a wide variety of tumors and cancers when conjugated with nanoparticles (NPs) and other nitric oxide (NO)-based compounds [9,12,13].

In this regard, NO donors, including S-nitrosothiols, have been largely employed in cancer therapy with great efficacy, through either their direct or indirect effects on several cellular biomolecules, such as proteins, lipids, and DNA [14]. It has also been demonstrated that NO effects are enhanced when NO donors are incorporated in nanoparticles against tumors and other pathogenic microorganisms [12,15,16,17,18]. This favors a sustained and gradual NO delivery into the target, thus mitigating cellular toxicity [19]. Moreover, there is an increased killing effect when NO is combined with other drugs and/or therapies, including nanomaterials [12,14,15,19].

Indeed, the potential of various types of nanomaterials has been also widely reported in almost every area of science. The broad range of its applications for medicine arises from its biomedical properties, making them suitable as antitumor and anticancer agents [20]. Particularly, metal nanoparticles or metal oxide nanoparticles have been largely explored in cancer therapy with promising results [20]. In particular, we have demonstrated that the combination of NO donor and metal-based NPs incorporated into biomaterials (such as polymeric films or hydrogels) can enhance toxicity against cancer cells [21,22]. Thus, the combination of metal-based NPs and NO donors in a biomaterial, such as a hydrogel, is a promising approach for localized and enhanced toxicity against cancer cells, with minimum side effects.

Herein, we propose an innovative topical approach for skin cancer. We demonstrate that PL combined with the NO donor GSNO and biogenic synthesized copper oxide nanoparticles (CuO NPs) have potent toxicity against melanoma cells. Both GSNO and CuO NPs were incorporated into PL hydrogel, allowing a topical application. We also evaluate the formulation potential as a therapy against melanoma cells and assess the possible underlying mechanisms of action. Thus, to give us some new insights into the interactions of the PL containing NO donor and CuO NP-based compounds with cancer cells, we investigate the mitochondrial membrane potential, cellular lipid peroxidation, and nuclear changes in treated cells.

## 2. Experimental Section

### 2.1. Green Synthesis and Characterization of CuO NPs

Green-tea-synthesized CuO NPs were prepared as previously described [23]. Briefly, an aqueous solution of CuCl2 (0.022 mM) was added dropwise into a suspension of green tea extract (2.5 mg·mL^−1^), prepared with ultrapure water, heated at 90 °C, and vacuum-filtered with a qualitative filter. The final suspension pH was adjusted to 5.4 using NaOH (1 M); then, the mixture was homogenized for 15 min with magnetic stirring. The final mixture was centrifuged and washed twice using ultrapure water.

This process led to the formation of a black precipitate of CuO NPs, which were freeze-dried and then stored in a desiccator protected from light. The structure of the CuO NPs was investigated using X-ray diffraction (XRD) measurements. Data were collected at a diffraction angle range between 30° and 80° at ambient temperature using a Bruker D8 diffractometer with Cu Kα1,2-radiation (λ = 0.154018 nm, 1600 W) source. The crystallite size was calculated according to the Debye–Scherrer equation (Equation (1)):(1)$$D=\frac{k}{\beta \;cos\;cos\;\theta \;}$$
where *D* is the diameter of the crystallite, λ is the wavelength for Cu kα, β is the full width at half maximum of the peak, θ is the Bragg diffraction angle, and *k* is a constant (0.94 is used to correspond spherical crystallites with cubic symmetry) [24].

Fourier-transform infrared (FTIR) spectroscopy analysis was recorded for the detection of functional groups on CuO NPs. Green tea-CuO NPs were placed in the FTIR sample holder on attenuated total reflectance mode (Spectrum Two, PerkinElmer, Waltham, MA, USA). Spectra were recorded from 700 to 4000 cm^−1^ at a 4 cm^−1^ resolution. The hydrodynamic size, polydispersity index (PDI), and zeta potential values of green-tea-synthesized CuO NPs were analyzed by dynamic light scattering (DLS, Zetasizer Nano ZS, Malvern Instruments Co, Worcestershire, UK).

### 2.2. Synthesis of GSNO

The NO donor, GSNO, was synthesized by reacting reduced *L*-glutathione (1.23 M) with an equimolar amount of sodium nitrite (NaNO_2_) in an aqueous HCl solution for 40 min in an ice bath, protected from light [12]. The obtained GSNO was precipitated by adding acetone, washed with cold water, vacuum-filtered, and freeze-dried for 48 h. Solid GSNO was maintained in a desiccator at −20 °C. The formation of GSNO was confirmed through detection of characteristic S-NO absorption bands at 336 nm (ε = 980.0 mol^−1^ L cm^−1^, associated with π → π* transition) or at 545 nm (ε = 18.4 mol^−1^ L cm^−1^, associated with nN → π* transition) which were measured using an Agilent 8454 UV–vis spectrophotometer (Palo Alto, CA, USA).

### 2.3. Preparation of PL Hydrogels Containing CuO NPs and/or GSNO

The hydrogels were prepared as previously reported with minor adaptations [11,25]. All hydrogels were prepared with 20% (*w/v*) Pluronic F-127. Thus, 1 g of solid Pluronic F-127 (PL) was added to 5 mL of PBS 1× and kept at 10 °C overnight, leading to the complete dissolution of the PL. This procedure led to the preparation of PL hydrogels (20% *w/v*).

For PL hydrogels containing CuO NPs, 5 mg of CuO NPs was added to the previously prepared PL hydrogel at 5 °C under gentle magnetic stirring. This process led to the formation of a PL hydrogel containing 0.1% (*w/v*) of CuO NPs, which will be referred to as PL + CuO.

For GSNO-containing PL hydrogels, 0.52 mg or 1.4 mg of GSNO was added to the previously prepared PL hydrogel at 5°C under gentle magnetic stirring. This process led to the formation of PL-GSNO hydrogel containing 312 µM GSNO or 624 µM, which will be referred to below as PL + GSNO.

For PL hydrogels containing CuO NPs and GSNO, 5 mg of CuO NPs and 0.52 mg or 1.4 mg of GSNO were added to the previously prepared PL hydrogel at 5 °C and under gentle magnetic stirring. This process led to the formation of a PL hydrogel containing 0.1% (*w/v*) CuO NPs and 312 µM GSNO or 624 µM, which will be referred to below as PL + CuO NPs + GSNO (Figure 1).

### 2.4. Transmission Electron Microscopy (TEM) of CuO NPs

CuO NP images were obtained using a Carl Zeiss LIBRA^®^ 120 transmission electron microscope (Zeiss International, Jena, Germany) operated at 80 kV. The sample was dispersed in isopropanol and submitted to an ultrasound bath for about 10 min. Then, a drop of this dispersion was deposited on a carbon grid. After solvent evaporation, the grid was placed in the desiccator and dried under vacuum for about 2 h. The average size of the nanoparticles in the solid state was determined using ImageJ software.

### 2.5. Scanning Electron Microscopy (SEM) and Energy-Dispersive X-ray Fluorescence Spectrometry (EDS) of the PL + CuO NPs + GSNO Hydrogels

The morphology of lyophilized PL + CuO NPs + GSNO hydrogel was examined with a field emission scanning electron microscope Quanta 250 (FEI Co, Hillsboro, OR, USA), equipped with an Oxford X-MAX50 energy-dispersive spectrometer (Oxford, UK), operated at 20 kV with a STEM (scanning transmission electron microscopy) detector. The dried samples were individually spread on double-sided conductive tape, placed in a metal stub, and then analyzed. The EDS technique was used to qualitatively map the distribution of carbon (C), oxygen (O), copper (Cu), and sulfur (S) atoms.

### 2.6. Kinetics of NO Release from PL Hydrogels

The kinetics of NO release from PL + GSNO and PL + CuO NPs + GSNO hydrogels were assessed by using a UV-vis spectrophotometer (Agilent 8454, Santa Clara, CA, USA) combined with a support of temperature control. The decay of absorbance intensity at 545 nm is associated with the cleavage of the S—N bond and the release of free NO from the GSNO. Thus, the absorbance intensity was monitored at λ = 545 nm, at 37 °C for 24 h. The amount of NO released as a function of time was returned from the initial concentration of GSNO, given by Equations (2) and (3). The experiment was performed in duplicate.
[NO]t = [GSNO]_0_ − [GSNO]t(2)
(3)$$[\mathrm{NO}]t=\frac{A_{0}l}{\varepsilon GSNO}-\frac{A_{t}l}{\varepsilon GSNO}$$

These equations relate the NO concentration at the time, where [GSNO]_0_ and [GSNO]t are the GSNO concentrations at the beginning of the reaction and at time t, respectively. Variables A_0_ and A_t_ are the GSNO absorbances at 545 nm, at the start of monitoring and at time t, respectively. The variable εGSNO is the molar absorption coefficient of GSNO at 545 nm (ε = 18.4 mol^−1^·L·cm^−1^), and l is the optical path of the cuvette, which corresponds to 1 cm [11].

### 2.7. In Vitro Diffusion of GSNO and Cu from PL Hydrogels

The in vitro kinetics of intact GSNO and Cu diffusion from the PL hydrogel were performed with a 7 mL vertical Franz diffusion cell (Hanson Research Corporation, Chatsworth, CA, USA) [21]. The cell consists of donor and receptor chambers that were separated with a hydrophilic nitrocellulose membrane with a 50 nm porosity and 25 mm diameter (Merck Millipore Ltd., Dublin, Ireland). The donor chamber was filled with 2 mL of PL hydrogel (at 30 mM of GSNO and 0.1% *w/v* of CuO NPs). A volume of 7 mL of PBS, pH 7.4, at 37 °C was added to the receptor chamber, with constant stirring. Every 30 min, a volume of 500 μL was withdrawn from the receptor chamber and replaced with the same volume of PBS. For the quantification of GSNO diffused from the hydrogels, the withdrawn samples were analyzed using UV−vis spectrophotometry by measuring the absorption band at 336 nm (π → π* transition, ε = 980 L·mol^−1^·cm^−1^), associated with S−N bond, as previously reported [11]. The amount of total copper diffused from the hydrogel through the membrane was measured with an inductively coupled plasma mass spectrometer (ICP-MS, Agilent 7900, Tokyo, Japan). For the ICP-MS analysis, 200 μL of aliquots of the withdrawn samples were diluted in 1.5 mL of an aqueous solution of 2% (*v/v*) nitric acid (HNO_3_) and the isotopes m/Z 63 and 65 were monitored.

### 2.8. Tumor Cell Culture and Cell Viability Assays

Murine B16F10 melanoma cells were harvested in RPMI 1640 medium (15 mM HEPES, 2 g of sodium bicarbonate/L, and 1 mM L-glutamine) and supplemented with 10% fetal bovine serum (FBS) at 37 °C and 5% CO_2_ atmosphere. Cells were seeded at a density of 5.10^3^ per well into a 96-well plate for 24 h. Then, the activity of PL hydrogels (alone or along with GSNO and/or CuO NPs) was assessed via the addition of serial dilutions of each compound at 4 different conditions: (1) PL, (2) PL + GSNO, (3) PL + CuO NPs and (4) PL + CuO NPs + GSNO, as outlined in Table 1. After 24 h, cellular viability was measured by incubating 10 μL of resazurin at 1.1 mg.mL^−1^ (Sigma-Aldrich, Burlington, MA, USA), for 4 h. Fluorescence intensity was evaluated through a microplate reader (Spectramax M4, Molecular Devices, San Jose, CA, USA) at λ_ex_ = 530 nm and λ_em_ = 590 nm. The results were normalized and expressed as a percentage of live cells. The half-maximal inhibitory concentration (IC_50_) of PL + CuO NPs + GSNO was calculated with a sigmoidal regression analysis using GraphPad Prism 7.0 software.

To further evaluate the efficacy of PL in combination with both CuO NPs and GSNO, a live/dead staining assay was conducted after treatment. Briefly, cells were treated with the top concentration of PL + CuO NPs + GSNO for 24 h, thus corresponding to 2.5 mg·mL^−1^, 100 µg·mL^−1^, and 625 µM of PL, CuO NPs, and GSNO, respectively. After treatment, cells were washed and stained with a live/dead kit assay (Sigma-Aldrich, Burlington, MA, USA) for 15 min according to the manufacturer’s instructions. Images were acquired with a fluorescence microscope (Nikon, Tokyo, Japan) and processed by ImageJ software [21].

### 2.9. PL + CuO NPs + GSNO Cytotoxicity on Fibroblasts

NIH 3T3 murine embryonic fibroblast cells were grown in DMEM medium (15 mM HEPES, 2 g of sodium bicarbonate/L, and 1 mM *L*-glutamine) and supplemented with 10% FBS at 37 °C and 5% CO_2_ atmosphere. In total, 5 × 10^3^ cells were allowed to adhere on 96-well plates for 24 h. Then, increasing concentrations (see Table 1) of PL + CuO NPs + GSNO were added to the wells and incubated for 24 h. Cellular viability was assessed by using resazurin at 1.1 mg·mL^−1^ (Sigma-Aldrich, USA) as described above and fluorescence intensity at λ_ex_ = 530 nm and λ_em_ = 590 nm was detected through a microplate reader (Spectramax M4, Molecular Devices, San Jose, CA, USA). Results were normalized and expressed as a percentage of live cells. The cytotoxic concentration (CC_50_) was calculated by sigmoidal regression analysis using GraphPad Prism 7.0 software.

### 2.10. Reactive Oxygen Species (ROS) and NO Detection Assays

To assess the total ROS production of (1) PL, (2) PL + GSNO, (3) PL + CuO NPs and (4) PL + CuO NPs + GSNO, cells were seeded at 5 × 10^3^ per well into a 96-well plate for 24 h and treated with the top concentration of each compound as shown in Table 1. Hydrogen peroxide (H_2_O_2_) at 20 mM was used as a positive control. Then, 10 µM of the fluorescent indicator 2′-7′-dichlorodihydrofluorescein diacetate (DCFH-DA) (Sigma-Aldrich, USA) was added to each well. Fluorescence intensity was measured every 2 min over a 5 h period using a microplate reader (Spectramax M4, Molecular Devices, San Jose, CA, USA) at λ_ex_ = 485 nm and λ_em_ = 535 nm.

To detect NO levels, melanoma cells were seeded and treated using the same method described above. Then, 10 µM of DAF-FM (4-amino-5-methylamino-2′,7′-difluorofluorescein) (ThermoFisher, Waltham, MA, USA) was added to each well. NO detection was carried out by taking reads every 5 min for 5 h at λ_ex_ = 495 nm and λ_em_ = 515 nm.

Fluorescence microscopy was also conducted to investigate the intracellular production of ROS and NO within melanoma cells after treatment. For this, 2 × 10^5^ tumor cells were grown onto round glass coverslips and allowed to attach to the bottom of 24-well plates, for 24 h. Cells were treated with PL-based hydrogels at the highest concentration of each compound for 3 h, as shown in Table 1. Cells were then washed and incubated with 10 µM of either DCFH-DA or DAF-FM for assessment of ROS (45 min) and NO (30 min), respectively. After that, cells were washed 3x with PBS, and images were obtained using a fluorescence microscope (Nikon, Tokyo, Japan) and processed by ImageJ software.

### 2.11. Nuclei Staining, Mitochondria Membrane Potential, and Lipid Peroxidation

To evaluate mitochondria membrane potential (ΔΨ_m_) and lipid peroxidation, fluorescence microscopy was performed to give us some insights into the mechanism of action of PL-hydrogel containing CuO NPs + GSNO. For this, glass coverslips were added to 24-well plates. Then, 2 × 10^5^ cells per well were seeded and treated with PL + CuO NPs + GSNO overnight. For mitochondrial labeling, cells were washed and incubated with 100 nM of MitoTracker CMXRos (Invitrogen, Waltham, MA, USA) for 45 min. Cells were fixed with methanol for 5 min and washed 3× with PBS. Nuclei were stained with DAPI at 2 μg·mL^−1^ for 10 min. Images were captured and fluorescence intensity was quantified in untreated control and treated cells using ImageJ software.

To assess changes in lipid metabolism, cells were seeded and treated as mentioned above. Then, 10 μM of Image-iT Lipid Peroxidation kit (ThermoFisher, Waltham, MA, USA) for live cells was added to each well for 30 min and washed 3× with PBS. Images were captured with a fluorescence microscope (Nikon, Tokyo, Japan). Lipid peroxidation was quantified using ImageJ software and the ratio of light intensity from the channels at 590 nm and 510 nm was calculated [22].

### 2.12. Statistical Analysis

Data are expressed as means ± standard deviation (SD) or standard error (SE) of the mean. When appropriate, statistical analysis was carried out by using two-way ANOVA followed by the Bonferroni post-test or using the unpaired *t*-test assessed by GraphPad Prism 7.0 software. The significance level was fixed at *p* < 0.05.

## 3. Results

### 3.1. CuO NP and PL Hydrogel Characterization

The crystalline structure of the CuO NPs is shown in Figure 2A. The resulting diffractogram shows a monoclinic structure of green-tea-synthesized CuO NPs, with peaks at 2θ (°) = 32.60, 35.59, 38.71, 48.93, 53.53, 58.27, 61.65, 66.42, 68.14, 72.52, and 75.10, corresponding to the ICSD number 016025. The crystallite size of the CuO NPs was calculated using Equation (1) and it was found to be 33.31 nm.

The FTIR technique was carried out to verify the presence of characteristic chemical bonds from phytochemicals derived from the green tea on the surface of CuO NPs, which act as reducing and capping agents (Figure 2B). The peaks at 3261.39 cm^−1^ and 2924.65 cm^−1^ are related to the O-H stretching and -CH vibration of aldehydes, respectively. These peaks might be associated with the OH groups of polyols such as catechins and CH groups of phenolic aldehydes [26,27,28]. Moreover, the peaks at 1582.42 cm^−1^, 1332.01, and 1017.09 cm^−1^ can be attributed to C=O groups of carboxylic acids, C-N stretching from caffeine (which is present in the composition of green tea), and C-O stretching, respectively [27,28].

Table 2 exhibits the hydrodynamic size, PDI, and surface charge of CuO NPs in an aqueous solution. The results indicate that there was a certain heterogeneity in the NPs and a stable colloidal dispersion due to the electrostatic repulsion evidenced by the zeta potential value. Overall, our data demonstrate that CuO NPs have a size at the nanoscale with moderate polydispersity and a negative zeta potential value, due to the presence of green tea polyphenol groups on the surface of the nanoparticles [28].

A representative electron micrograph obtained using TEM is shown in Figure 3A. As can be observed, the nanoparticles had a spherical shape, and their average diameter size was 22.94 ± 0.82 nm (Figure 3B). It is also possible to note that the nanoparticles were well dispersed.

EDS microanalysis qualitatively identified the elemental composition of the hydrogel (Figure 4). Through the map, it is possible to observe that there was a higher density of carbon and oxygen atoms as expected, since these are the basis atoms of Pluronic F-127. Sulfur was also identified due to its presence in the GSNO and copper was identified due to the presence of CuO NPs. Importantly, all mapped atoms were well dispersed without the formation of microdomains in the hydrogel.

The diffusion of intact GSNO and Cu from PL + CuO NPs + GSNO hydrogel was monitored using the Franz diffusion cell, as shown in Figure 5 [21,22]. The diffusion profile of intact GSNO molecules from the hydrogel shows an initial burst in the first 2 h of monitoring, followed by a steady-state GSNO diffusion (Figure 5A). Indeed, the diffusion rate for GSNO in the first 2 h was found to be 0.83 mM·h^−1^, and from 2.5 to 6 h, it was 0.15 mM·h^−1^. By calculating the area under the curve, the total amount of GSNO diffused from the hydrogel was 11.19 mM, after 6 h of monitoring. Copper presents a similar diffusion profile, compared to GSNO (Figure 5B). In the first 2 h of monitoring, the diffusion rate for copper was 8.64 µg·mL^−1^·h^−1^ and then it became 2.81 µg·mL^−1^·h^−1^ from 2.5 to 6 h. The total amount of copper released during 6 h of monitoring was 113.79 µg·mL^−1^. Similar results were reported for GSNO diffusion from polymeric hydrogels [22]. Thus, these results indicate that PL hydrogel can promote a sustained diffusion of both GSNO and copper. Yet, PL hydrogel is able to release both the intact GSNO molecule and copper in a similar fashion.

Figure 6 shows the NO release profile from GSNO. Once synthesized, GSNO underwent spontaneous thermal decomposition due to the homolytic S-N cleavage releasing free NO, producing oxidized glutathione (GSSG), as shown in Equation (4) [29].
2 GSNO → 2 NO + GSSG(4)

As can be observed, both hydrogels released NO in a sustained manner, for at least 24 h. The PL + GSNO hydrogel released a total amount of approximately 3.5 mM of NO. Although there was no difference in the total amount of NO release, the addition of CuO NPs changed the NO release profile, where the hydrogel without CuO NPs had a linear profile, while the hydrogel containing CuO NPs increased linearly until 5 h and then slowed down before reaching a plateau. Particularly, at the beginning of the monitoring, there was a 2-fold increase in the initial rates of NO release with the addition of CuO NPs, from 0.14 ± 0.009 mM·h^−1^ to 0.34 ± 0.01 mM·h^−1^.

### 3.2. Cytotoxicity of PL-Based Hydrogels against Melanoma Cells

To determine the activity of all PL-based hydrogels, melanoma cells were exposed to increasing concentrations of each compound for 24 h. As a result, no significant differences were found between the untreated cells and those treated only with PL, thus suggesting that the hydrogel matrix was not toxic regardless of the concentration used. However, when cells were challenged with the other combinations, significant differences were found. Cellular viability decreased by 37.6% and 41.7% when the cells were treated with PL + GSNO (0.62 mg·mL^−1^ + 156 μM) and PL + CuO NPs (0.62 mg·mL^−1^ + 25 μg·mL^−1^), respectively. No statistically significant differences were observed between either group. The killing effect was more pronounced when all three compounds were associated. As shown in Figure 7A, PL + GSNO + CuO NPs (0.62 mg·mL^−1^ + 156 μM + 25 μg.mL^−1^) reduced the percentage of live cells by 60.8%.

The increase in drug concentration resulted in a better killing rate. At a higher concentration, PL + GSNO (2.5 mg·mL^−1^ + 625 μM) reduced cellular viability by 56.2 %, while PL + CuO NPs (2.5 mg·mL^−1^ + 100 μg.mL^−1^) caused a greater response, thus killing 84.7 % of tumor cells with statistically significant differences between both. The combination of PL + GSNO + CuO NPs (2.5 mg·mL^−1^ + 625 μM + 100 μg·mL^−1^) substantially decreased the number of viable cells (in about 93.4%), hence suggesting a synergistic effect occurred from the interaction between GSNO and CuO NPs placed into the hydrogel matrix.

We also demonstrated that when the cells were challenged with the highest concentration of PL + GSNO + CuO NPs, nearly all the cells were dead (red fluorescence) 24 h post-treatment, while in the untreated control, most cells were alive (green fluorescence), hence confirming the great potential of this compound (Figure 7B).

### 3.3. The Assessment of IC_50_ and CC_50_ Values of PL + CuO NPs + GSNO on Melanoma and Fibroblast Cells

By using sigmoidal regression analysis, we calculated the IC_50_ value of 18.1 μg.mL^−1^ for CuO NPs and 113.3 μM for GSNO when melanoma cells were treated at different concentrations of PL + CuO NPs + GSNO (Figure 8A,C). Moreover, the cytotoxicity assay of PL hydrogel containing GSNO and CuO NPs was conducted to determine the effects of this drug on healthy skin cells (NIH3T3). Interestingly, the cytotoxic concentration (CC_50_) of both GSNO and CuO NPs against fibroblasts was 2-fold higher than that obtained for cancer cells, thus resulting in a concentration of 37 μg·mL^−1^ and 231.3 μM for CuO NPs and GSNO, respectively (Figure 8B,D). This resulted in a selectivity index (SI) of nearly 2, meaning that PL + CuO NPs + GSNO had greater activity toward cancer cells than healthy cells. SI was determined according to Equation (5) [30].
(5)$$SI=\frac{CC50}{IC50}$$

### 3.4. ROS and NO Production on Melanoma Cells

Our results show that the untreated cancer cells produced steady amounts of ROS over time, which increased 2-fold when treated only with PL. When the cells were treated with H_2_O_2_ at 20 mM, we observed a significant enhancement in the ROS level (about 13-fold) in the first 60 min, followed by a steady-state condition. This was the lowest H_2_O_2_ concentration capable of producing ROS. However, in cells treated with PL + GSNO, there was a 14-fold increase in the levels of ROS after 60 min compared to the control. The levels of ROS were greatly increased in the group exposed to PL + CuO NPs, thus resulting in amounts nearly 31 times higher than those found in untreated cells.

Remarkably, the combination of all three compounds caused an overproduction of ROS, resulting in a 49-fold increase compared to the control. In addition, by calculating the area under the curve, we determined that PL + CuO NPs + GSNO produced around 1.5-fold and 3.4-fold more ROS than PL + CuO NPs and PL + GSNO over time, respectively. Importantly, the steady-state condition in ROS release only appeared after a 5 h period of drug incubation in groups treated with PL + GSNO, PL + CuO, and PL + CuO NPs + GSNO (Figure 9A).

These results were further confirmed using fluorescence microscopy, which showed that the levels of ROS were substantially higher in the cells treated with PL + GSNO, PL + CuO, and PL + CuO NPs + GSNO, represented by the greater intensity of green fluorescence in those cells when compared to the other controls (untreated, PL, and H_2_O_2_) (Figure 9B).

In terms of NO production, low amounts of NO were detected in the untreated control and the cells treated with PL and PL + CuO NPs (Figure 10A,B(a–c)). In contrast, NO levels were substantially enhanced (by 6.2-fold) when treated with PL + CuO NPs + GSNO in the first 30 min after incubation. After 1 h, the levels of NO reached a steady state, which was sustained for at least 5 h post-treatment. Remarkably, the levels of NO were tremendously improved by the treatment with PL + GSNO, achieving values nearly 19-fold higher than the control after 3 h of incubation, which was also sustained for at least 5 h (Figure 10A). We also demonstrated that in both groups containing GSNO, the green fluorescence intensity was significantly higher than that shown in the other groups, thus suggesting the treatment produced great amounts of NO within the cells in a very short period (Figure 10B(d,e).

### 3.5. The Assessment of Nuclear Changes, Mitochondrial Membrane Potential, and Lipid Peroxidation

Nuclear staining suggests that most treated cells exhibited chromatin condensation, as defined by the very bright blue fluorescence intensity in Figure 10A. There was a considerable loss of ΔΨm in treated cells, determined by the lower intensity in the red fluorescence compared to the control (Figure 11A). In addition, we found that ΔΨm decreased by 28%, hence suggesting our compound produced a significant impact on the depolarization of mitochondrial membranes of melanoma cells (Figure 11C).

Moreover, there was a significant increase in lipid peroxidation on treated cells, as indicated by the high intensity of green fluorescence. As shown in Figure 11B, treated cells presented huge amounts of oxidized lipids as opposed to the fewer number of reduced lipids (represented by the low intensity in red fluorescence). The ratio between reduced/oxidized lipids was tremendously reduced (by nearly 40%), thus suggesting PL + CuO NPs + GSNO also produced a significant effect on the lipid metabolism of tumor cells (Figure 11D).

## 4. Discussion

In this work, we successfully synthesized and characterized PL hydrogels combined with the NO donor GSNO and green-tea-synthesized CuO NPs. We also demonstrated its potential to kill malignant melanoma cells in a concentration-dependent manner.

The crystalline structure of CuO NPs was identified through XRD analysis and peaks referring to the intrinsic bonds of CuO NPs and the presence of green tea-derived phytochemicals on the surface of CuO NPs were identified by FTIR. Also, the hydrodynamic size and zeta potential analysis indicated that the nanoparticles are following the values reported in the literature and form stable colloidal suspensions [31]. Through TEM it was possible to observe the morphology of the nanoparticles, which proved to be spherical and well dispersed. Besides that, the mapping of carbon, oxygen, copper, and sulfur of the PL + CuO NPs + GSNO hydrogel was performed by using the EDS technique. The atomic mappings showed a homogeneous distribution of the mapped atoms, without the formation of microdomains.

Regarding the diffusion assays, it was noticed both the intact GSNO molecule and copper have a sustained diffusion from the PL hydrogel. During 6 h of monitoring, approximately 37% of the total amount of GSNO was diffused from the hydrogel, while only 10% of copper in the ionic form was diffused, indicating that these nanoparticles are stable. Besides that, the diffusion profiles for both compounds were similar, since it was possible to see a linear behavior at the beginning, and after a couple of hours, they reached a steady state (see Figure 5).

Moreover, the NO release kinetics from both PL + GSNO and PL + CuO NPs + GSNO hydrogels were in the mM range (responsible for antitumor effects) for at least 24 h. Interestingly, the addition of copper to PL + GSNO hydrogel seemed to improve the release of NO in the first hours of the reaction, evidenced by the increase in the NO release rate in the first 3 h, doubling the NO release rate. Copper is a well-known catalytic agent and the trace of Cu^+2^ ions from nanoparticles probably increased the disruption rate of the S-NO bond [32]. Then, when these ions saturated, there was a decrease in the NO release, as evidenced by the steady state (see Figure 6).

Our results show that PL alone is not toxic to either cancer or healthy cells and, thus, is an attractive carrier system for drug delivery. Indeed, PL-based compounds have been shown to improve drug biodistribution. It has been demonstrated that the antitumor effects of the chemotherapy drug doxorubicin were enhanced when conjugated with Pluronic formulations, which was given by the increase in drug stability and solubility on the PL matrix when compared with the free drug [33]. In addition, because of their adhesive and biocompatible properties, PL-based hydrogels are suitable for wound covering, being ideal formulations for topical administration on injured tissue [9].

PL hydrogels have been largely applied in combination with NO donors to allow a sustained NO release against tumor cells and several pathogens [11,13]. Here, the association of PL with GSNO (PL + GSNO) resulted in a significant reduction in melanoma cell viability at the highest tested concentrations. Indeed, the dual role of NO on various tumor cells has been widely reported [14,34]. Although NO was assumed to be an oncogenic molecule for many years, recent studies have shown that NO can also have inhibitory effects at high concentrations, thus leading to apoptosis, DNA damage, and tumor gene suppression [14]. Moreover, besides ensuring controlled NO release, the activity of GSNO on the cellular environment is also of critical importance, presenting numerous therapeutic applications [35]. One of the major actions of GSNO comprises the S-nitrosylation and transnitrosation process of cysteine protein thiols, resulting in perturbations in cellular functions and therefore affecting its homeostasis and leading to cell death [35,36].

However, when PL was combined with CuO NPs (PL + CuO NPs), the killing rate against melanoma cells was even more pronounced. CuO NPs have been employed for several biomedical applications with impressive antimicrobial and antitumor activities [37,38]. In particular, those NPs have shown great potential against different lines of breast cancer as well as lung, colon, and cervical cancer due to the increase in ROS production [38]. Additionally, cuprous oxide (Cu_2_O), another type of copper oxide, showed a significant impact against melanoma in vitro and in vivo, improving the survival rate of tumor-bearing mice and also preventing metastasis [39].

Therefore, considering the individual potency of each drug, the combination of both GSNO and CuO NPs incorporated into the PL matrix (PL + CuO NPs + GSNO) resulted in a significant synergistic effect, producing a tremendous impact on the killing of cancer cells. Moreover, it showed a greater anticancer activity, being more selective for melanoma rather than healthy fibroblast cells, which is essential for a treatment to be considered effective. Indeed, the combination resulted in an SI of about 2, which was higher than one of the most used chemotherapy drugs (dacarbazine) to treat advanced stages of skin cancer, which showed an SI of about 1 against melanoma cells [30]. Yet, it has been reported that the SI of other two well-known chemotherapeutic agents (cisplatin and doxorubicin) was equivalent to dacarbazine (approximately 1) against melanoma and breast cancer [30,40]. A recent study also reported a low SI for doxorubicin (nearly 1.6) against a prostate cancer cell line [41].

Thus, we assume that there were some reasons behind the achievement of our remarkable results. Initially, the levels of ROS were dramatically increased by the incorporation of CuO NPs and GSNO into the PL matrix. It is known that ROS can affect the DNA by inducing double-strand breaks and oxidizing nucleoside bases, such as guanine and adenine, therefore leading to chromosomal rearrangements and condensation, triggering programmed cell death [42,43]. In addition, ROS can cause changes in mitochondrial DNA, which is degraded by a severe oxidative stress condition [43].

The overproduction of ROS can also lead to a loss of ΔΨm resulting in mitochondrial dysfunction and cell death by apoptotic pathways. Indeed, mitochondria are the “powerhouses” of cells, thus being essential for ATP production and cell survival [44]. Therefore, targeting mitochondria as an anticancer therapeutic strategy has been a field of considerable interest [44].

ROS-induced lipid peroxidation was also a crucial point that contributed to our excellent results. It has been demonstrated that high levels of ROS can attack and impair the lipid bilayer of cellular membranes, which are essentially made of polyunsaturated fatty acids (PUFAs) [45]. These are major components of biological membranes that are highly susceptible to oxidative damage [45]. As a result, membranes’ fluidity, permeability, and stability are seriously compromised [45,46]. Furthermore, lipid peroxidation can also induce a chain reaction in which the end products, such as aldehydes (mainly malondialdehyde and 4-hydroxynonenal), can react with multiple biomolecules, thiols, amino groups, as well as DNA, with this being potentially lethal to the cells [46,47].

The high amounts of NO produced were another key factor contributing to the success of our treatment. Although the group receiving only PL + GSNO presented greater levels of NO, PL + CuO NPs + GSNO also generated increased amounts compared to the other treated groups. It is worth mentioning that the NO release was sustained for at least 5 h.

NO can directly influence several biomolecules through protein nitration, inflicting unusual signaling responses [35]. Moreover, it can react with the superoxide radical in biological systems, hence producing peroxynitrite (ONOO^−^), which is a highly reactive short-lived oxidative species [10]. Thus, we assume that the lower levels of NO detected in cells treated with PL + CuO NPs + GSNO were a result of the reaction that might have occurred between NO and other ROS preventing the formation of free NO.

ONOO^−^ can directly or indirectly cause a significant impact on multiple organelles, also affecting DNA, mitochondria, proteins, and lipids [48]. Of note, PUFAs are also oxidized by this reactive nitrogen species producing new oxidized lipids containing nitrogen [10]. In fact, several of the biological effects of NO have been attributed to the formation of ONOO^−^ [48]. For example, ONOO^−^ has been responsible for the mitochondrial inactivation of cells by inhibiting electron transport [48]. NO can inhibit cytochrome c oxidase, thus allowing superoxide anion leakage and the production of ONOO^−^, which is tremendously deleterious to mitochondrial function [48]. In addition, it can easily diffuse across cellular membranes, hence affecting the neighboring cells very quickly [10,48]. Thus, because of the increased formation of such reactive species, we suggest that our outstanding results were enhanced by the association of all compounds.

CuO NPs are also known to generate oxidative stress in cells since they act as a catalyst and increase the formation of ROS through Fenton reactions or Haber–Weiss reactions, which produce hydroxyl radicals. These nanoparticles interact with the membrane and are transported into the cell via endocytosis, initiating the generation of intracellular ROS by catalyzing the reaction of free radicals in the mitochondria. Thus, CuO nanoparticles block the electron transport chain and increase oxygen free radicals that lead to oxidative stress. This stress can generate apoptosis, DNA damage, cytotoxicity, dysregulated cell signaling, etc. Thus, the combination of CuO NPs with the NO donor generated a powerful material that induces reactive species and, consequently, has potent antitumor activity [49].

In summary, in this work, we demonstrate that PL-based hydrogels containing either GSNO or CuO NPs have good potential to kill cancer cells. However, the efficacy of our therapy against melanoma was tremendously potentiated by the combination of GSNO and CuO NPs in the polymeric matrix. Moreover, PL + CuO NPs + GSNO significantly increased the levels of ROS and NO, triggering multiple cellular responses in different organelles. Most importantly, it was preferentially selective for melanoma rather than healthy cells. This opens a new avenue for the treatment of skin cancer. Our promising results show the great potential of PL + CuO NPs + GSNO to overcome the current challenges or undesired outcomes of melanoma chemotherapy.

## Figures and Tables

**Figure 1 pharmaceutics-15-01971-f001:**
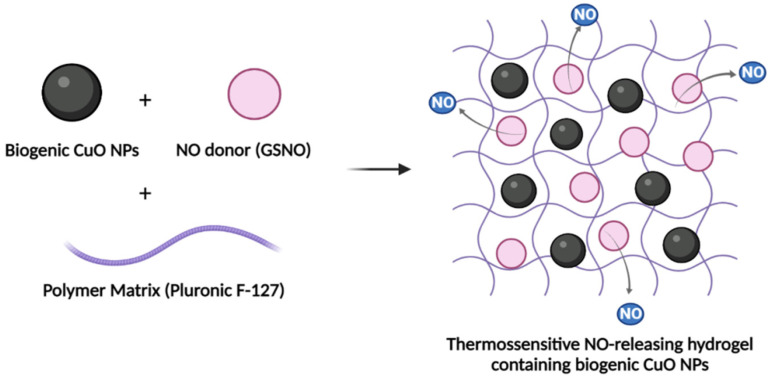
Schematic representation of the formation of PL hydrogel containing both NO donor (GSNO) and CuO NPs for toxicity against cancer cells.

**Figure 2 pharmaceutics-15-01971-f002:**
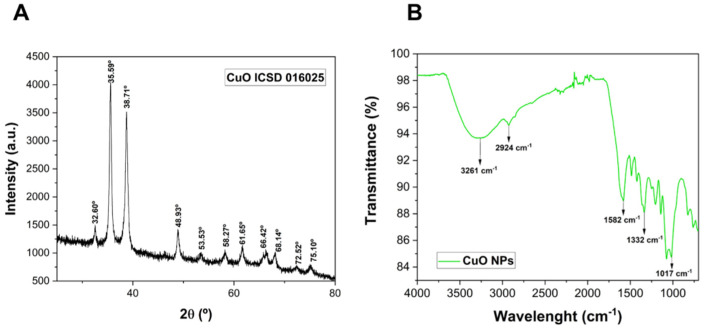
(**A**) Diffractogram of CuO NPs and (**B**) FTIR spectrum of CuO NPs.

**Figure 3 pharmaceutics-15-01971-f003:**
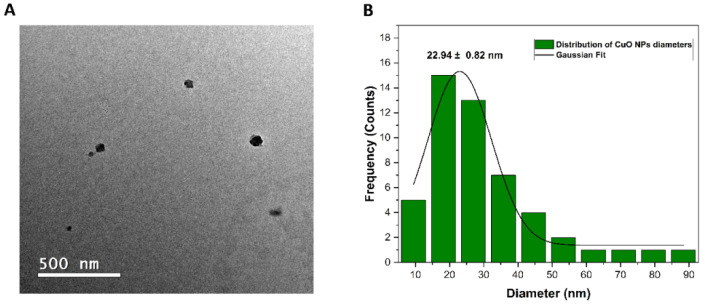
Representative TEM image (**A**) and particle size distribution chart (**B**) of CuO NPs.

**Figure 4 pharmaceutics-15-01971-f004:**
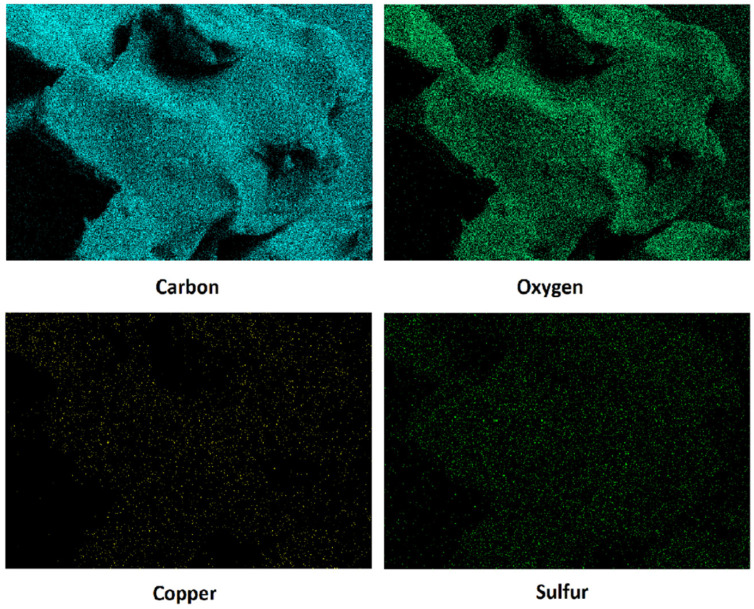
EDS mapping for carbon, oxygen, copper, and sulfur atoms in PL + CuO NPs + GSNO hydrogels.

**Figure 5 pharmaceutics-15-01971-f005:**
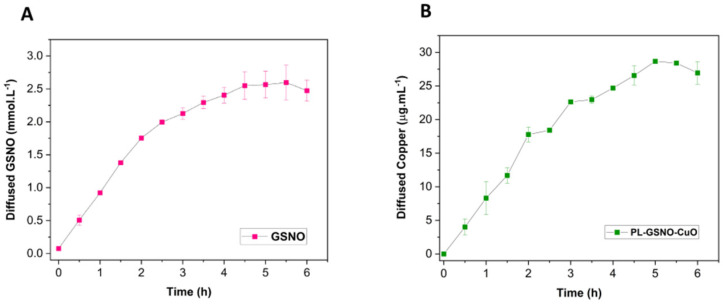
Franz cell diffusion profile of (**A**) GSNO and (**B**) copper from PL + CuO NPs + GSNO hydrogels for 6 h, at 37 °C. The results are reported as mean ± SD of two independent experiments.

**Figure 6 pharmaceutics-15-01971-f006:**
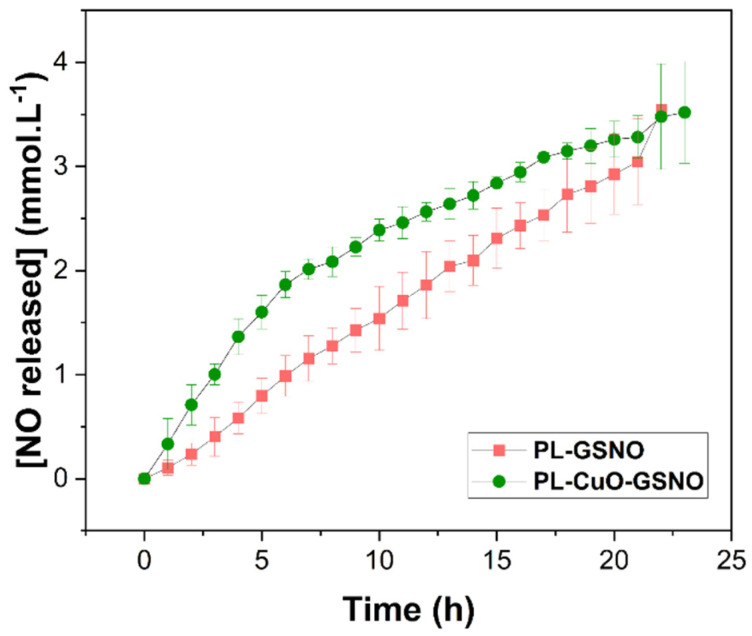
Kinetics of NO release from PL + GSNO (pink line) and PL + CuO NPs + GSNO (green line) for 24 h at 37 °C. The results are reported as means ± SD of two independent experiments.

**Figure 7 pharmaceutics-15-01971-f007:**
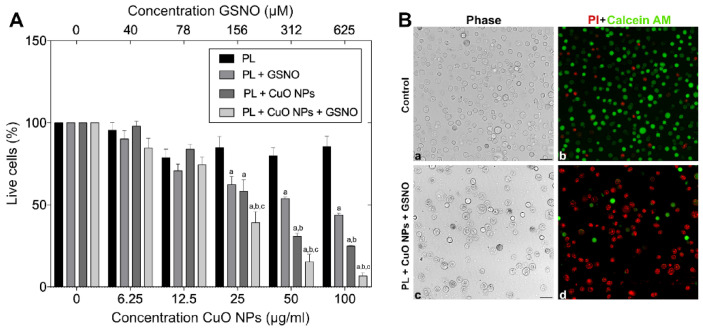
(**A**) Mean values ± SE of the activity of PL, PL + GSNO, PL + CuO NPs, and PL + CuO NPs + GSNO on melanoma cells 24 h post-treatment. ^a^ denotes statistically significant differences from PL, ^b^ denotes statistically significant differences from PL+GSNO, and ^c^ denotes statistically significant differences from PL + CuO NPs. (**B**) Live/dead staining assay 24 h after treatment for control (a, b) and treated cells (c, d). Green fluorescence represents live cells. Red fluorescence represents dead cells. Original magnification: 200×. Scale bar = 100 μm.

**Figure 8 pharmaceutics-15-01971-f008:**
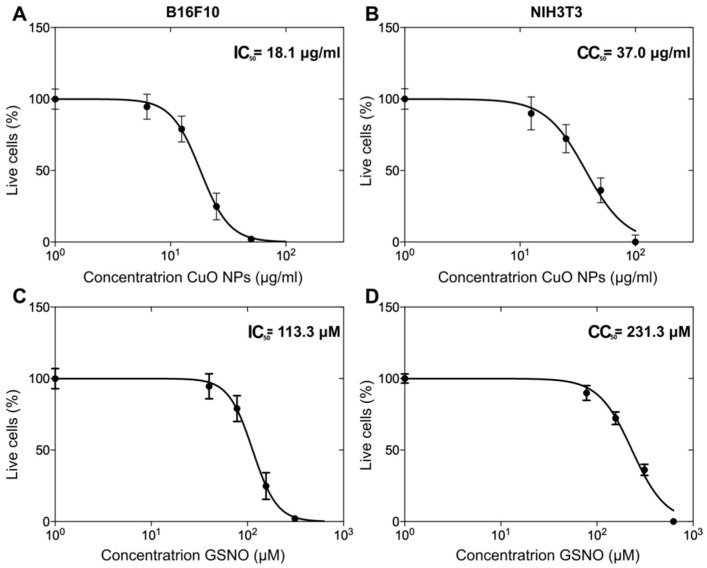
Dose–response curve of CuO NPs for melanoma (**A**) and fibroblasts (**B**) and GSNO for melanoma (**C**) and fibroblasts (**D**). Data are plotted as mean values ± SE. IC_50_ (tumor cells) and CC_50_ (healthy cells) are highlighted.

**Figure 9 pharmaceutics-15-01971-f009:**
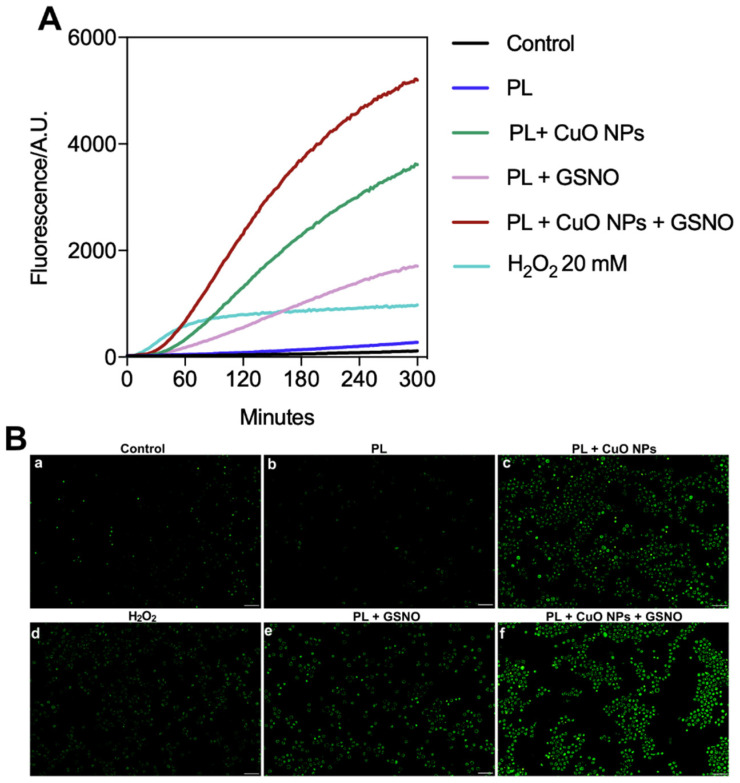
(**A**) ROS detection in melanoma cells after treatment with PL-based hydrogels over time. (**B**) Fluorescence microscopy images of melanoma cells using the indicator dye DCFH-DA 3 h after treatment. (**a**) Untreated control, (**b**) PL, (**c**) PL + CuO, (**d**) H_2_O_2_ (**e**) PL + GSNO, and (**f**) PL + CuO NPs + GSNO. Original magnification: 200×. Scale bar = 100 μm.

**Figure 10 pharmaceutics-15-01971-f010:**
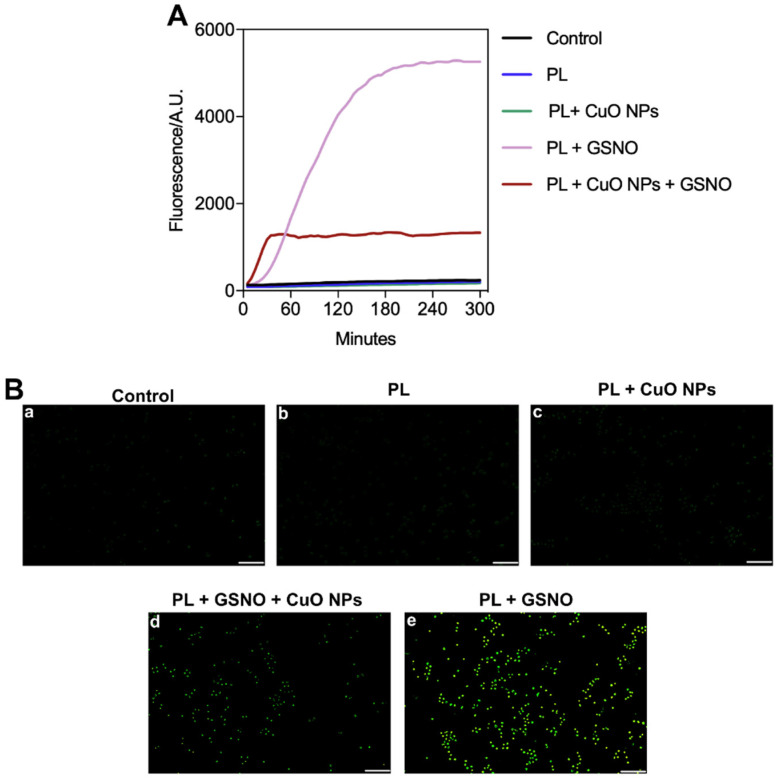
(**A**) NO detection in melanoma cells after treatment with PL-based hydrogels over time. (**B**) Fluorescence microscopy images of melanoma cells using the indicator dye DAF-FM 3 h after treatment. (**a**) Untreated control, (**b**) PL, (**c**) PL + CuO, (**d**) PL + CuO NPs + GSNO and (**e**) PL + GSNO. Scale bar = 100 μm.

**Figure 11 pharmaceutics-15-01971-f011:**
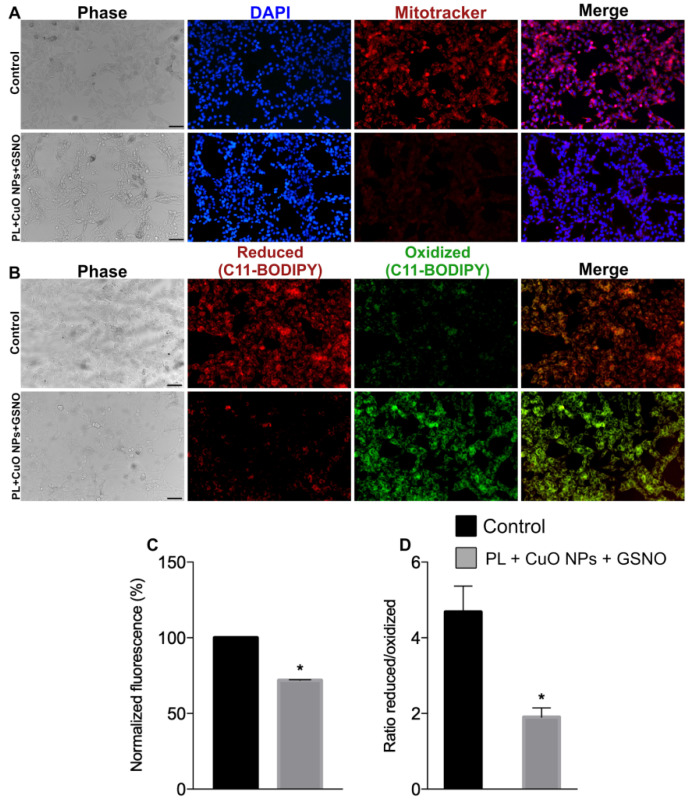
(**A**,**C**) Mitochondrial membrane potential and (**B**,**D**) lipid peroxidation of melanoma cells treated with PL + CuO NPs + GSNO. Values shown represent the means ± SD. * denotes statistically significant differences between control and treated tumor cells. Scale bar = 100 µm.

**Table 1 pharmaceutics-15-01971-t001:** Concentrations of Pluronic F-127 hydrogel, CuO NPs, and GSNO on melanoma cells. These concentrations were established based on preliminary assays in which we obtained synergy between CuO NPs and GSNO.

Pluronic F-127 (mg·mL^−1^)	CuO NPs (µg·mL^−1^)	GSNO (µM)
0.15	6.25	40
0.31	12.5	78
0.62	25	156
1.25	50	312
2.5	100	625

**Table 2 pharmaceutics-15-01971-t002:** Hydrodynamic size, PDI, and ζ potential measurements of CuO NPs.

Hydrodynamic Size ± SD (nm)	PDI ± SD	ζ Potential ± SD (mV)
250 ± 64.22	0.382 ± 0.05	−14.53 ± 0.47

## Data Availability

The corresponding authors can provide the data upon reasonable request.

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
