# Peer review of "Pluronic F-127 Hydrogels Containing Copper Oxide Nanoparticles and a Nitric Oxide Donor to Treat Skin Cancer"

_pharmaceutics, 2023, doi:10.3390/pharmaceutics15071971_

Round 1

Reviewer 1 Report

In the present study, the authors report the potential effect of Pluronic F-127 hydrogels containing copper oxide nanoparticles and a nitric oxide donor on skin cancer cells. In summary, the manuscript presents valid evidence of the potential effect. However, I would like to discuss the following comments with the authors.

1) L82: “as shown in Figure 1” I advise to move figure 1 to the appropriate location in the methodology or the results.

2) The authors need to justify the usage of these particular concentrations used for the cell viability assays and listed in table 1

3) L200 and L221: why the authors decided to use the term “EC50” and “CC50” instead of “IC50”?

4) The authors should include a “Particle size distribution chart”

5)The TEM image of CuO NPs raises a question about the amount of the biosynthesized NPs.

6)Figure 7B: magnification should be stated as well in the caption

7)L387: the authors should state a reference for Equation 5

8)Fig 9A: why the authors decide to use this specific concentration for H2O2 (20 mM)

9)Figure 9B: magnification should be stated as well in the caption

10)How the authors can explain/relate why the “PL + GSNO” showed the maximum level of NO production although the “PL + CuO NPs + 420 GSNO” showed the maximum effect in terms of ROS detection.?

Author Response

Dear reviewer #1,

Thank you very much for your time and consideration of our manuscript. Below you will find our responses to your comments. All changes are highlighted in yellow in the manuscript.

In the present study, the authors report the potential effect of Pluronic F-127 hydrogels containing copper oxide nanoparticles and a nitric oxide donor on skin cancer cells. In summary, the manuscript presents valid evidence of the potential effect. However, I would like to discuss the following comments with the authors.

1) L82: “as shown in Figure 1” I advise to move figure 1 to the appropriate location in the methodology or the results.

We moved Figure 1 to the methodology, thank you for your suggestion.

2) The authors need to justify the usage of these particular concentrations used for the cell viability assays and listed in table 1.

The concentrations were selected based on preliminary assays in which we obtained synergy between CuO NPs and GSNO. We addressed this issue in the manuscript.

3) L200 and L221: why the authors decided to use the term “EC50” and “CC50” instead of “IC50”?

The reviewer is right and we apologize for that. We changed EC50 to IC50.

4) The authors should include a “Particle size distribution chart”

We included a “particle size distribution chart” (Figure 3B) as suggested by the reviewer.

5) The TEM image of CuO NPs raises a question about the amount of the biosynthesized NPs.

The TEM image of CuO NPs is a representative image of the nanoparticles. To obtain the images, the nanoparticle dispersion was significantly diluted. Thus, we obtained a low population of the nanoparticles in order to better observe their morphology. We addressed this issue in the manuscript.

6) Figure 7B: magnification should be stated as well in the caption

We added the magnification as per your request.

7)L387: the authors should state a reference for Equation 5

We added a reference as per your request.

8)Fig 9A: why the authors decide to use this specific concentration for H2O2 (20 mM)

Indeed, we tested different concentrations of H2O2 and 20 mM was the lowest concentration capable of producing ROS. We addressed this issue in the manuscript.

9)Figure 9B: magnification should be stated as well in the caption

We added magnification as per your request.

10 )How the authors can explain/relate why the “PL + GSNO” showed the maximum level of NO production although the “PL + CuO NPs + GSNO” showed the maximum effect in terms of ROS detection?

Indeed, NO can react with the superoxide radical in biological systems, hence producing peroxynitrite (ONOO-), which is a highly reactive short-lived oxidative species. Thus, we assume that lower levels of NO detected in cells treated with PL + CuO NPs + GSNO were a result of the reaction that might have occurred between NO and other ROS preventing the formation of free NO. We addressed this issue in the manuscript.

Reviewer 2 Report

The subject of the manuscript entitled “Pluronic F-127 hydrogels containing copper oxide nanoparticles 2 and a nitric oxide donor to treat skin cancer” is interesting, overall well written, comprehensive presented and it faces an interesting issue with possible biomedical applications. Here following the comments/remarks on specific issues the Authors should address and that could improve their work:

-Experimental Section: Characterization of synthesized GSNO should be presented (Suppl.?)

-Results: ROS and NO production on melanoma cells. What about the effect on healthy cells, if any?

-Results: Nuclear changes, mitochondrial membrane potential and lipid peroxidation. What about the effect on healthy cells, if any?

Results: the effect on melanoma and fibroblasts cell cycle was checked?

In conclusion the manuscript fits with the scope of the journal and the authors have done a good work finally. However, based on my comments the manuscript can be published after explanations required above.

Correct  English in general ! Only minor editing of language required.

Author Response

Dear reviewer #2,

Thank you very much for your time and consideration of our manuscript. Below you will find our responses to your comments. All changes are highlighted in yellow in the manuscript.

The subject of the manuscript entitled “Pluronic F-127 hydrogels containing copper oxide nanoparticles 2 and a nitric oxide donor to treat skin cancer” is interesting, overall well written, comprehensive presented and it faces an interesting issue with possible biomedical applications. Here following the comments/remarks on specific issues the Authors should address and that could improve their work:

-Experimental Section: Characterization of synthesized GSNO should be presented (Suppl.?)

GSNO has two absorption bands at 336 nm (associated with π ® π* transition) and at 545 nm (associated with nN ® π* transition), both bands are associated with S-NO group. Once synthesized, GSNO was characterized by its Uv-vis absorption bands. As our group has previously reported this characterization in several published papers (J. Phys. Chem. A 2002, 106, 8963-8970;  Biomaterials 24 (2003) 3543–3553; Biomaterials 25 (2004) 3773–3782; Biomacromolecules 2005, 6, 2512-2520; Journal of Pharmaceutical Sciences 94 (2005) 994–1003; Journal of Drug Delivery Science and Technology 43 (2018) 211–220) we prefer not include a Figure in the Supplementary material (it would be just only one Figure). However, we addressed this issue in the manuscript.

-Results: ROS and NO production on melanoma cells. What about the effect on healthy cells, if any?

Indeed, we measured ROS and NO levels only for melanoma cells since the selectivity index was approximately 2. In this case, it would be expected much lower ROS and NO levels for fibroblasts.

-Results: Nuclear changes, mitochondrial membrane potential and lipid peroxidation. What about the effect on healthy cells, if any?

We verified nuclear changes, mitochondrial membrane potential, and lipid peroxidation as death pathways for melanoma cells. As previously reported, we used conditions in which the selectivity index was around 2. In these conditions, it is not expected damage on fibroblasts.

-Results: the effect on melanoma and fibroblasts cell cycle was checked?

In this work, we did not check cell cycle for tumor and healthy cells. Indeed, our purpose was not to compare the effects of PL-CuONP-GSNO between melanoma and fibroblasts. We aimed to develop pluronic F-17 hydrogels with CuONPs and GSNO against melanoma. We identified a condition that can kill melanoma cells but not fibroblasts. In this case, we think cell cycle assays do not add further information to the manuscript.

In conclusion the manuscript fits with the scope of the journal and the authors have done a good work finally. However, based on my comments the manuscript can be published after explanations required above.

Thank you for your kind words. We hope we addressed all of your concerns.

Round 2

Reviewer 2 Report

The manuscript entitle “Pluronic F-127 hydrogels containing copper oxide nanoparticles 2 and a nitric oxide donor to treat skin cancer” was revised and the version 2 revealed an improved form.

I have one suggestion:

- To insert the answers given to my questions in Discussion Section

And some comments:

-cell cycle analysis provides useful information regarding the mode of action against melanoma vs normal cells;

-in the case of drugs it is always necessary to have as control normal, healthy cells. Not to assume, but to demonstrate!

In conclusion, the manuscript could be published in the presented revised form.